# Palladium-catalyzed difluorocarbene transfer enables access to enantioenriched chiral spirooxindoles

Zhiwen Nie[1,2,5], Keqin Wu[1,3,5], Xiaohang Zhan[2,5], Weiran Yang [ID][3], Zhong Lian [ID][4], Shaoquan Lin[1], Shou-Guo Wang [ID][2] & Qin Yin [ID][1,2] ✉

We disclose herein an unprecedented Pd-catalyzed difluorocarbene transfer reaction, which assembles a series of structurally interesting chiral spiro ketones with generally over 90% ee. Commercially available $BrCF_2CO_2K$ serves as the difluorocarbene precursor, which is harnessed as a user-friendly and safe carbonyl source in this transformation. Preliminary mechanistic studies exclude the formation of free CO in the reaction process, and importantly, we also find that $BrCF_2CO_2K$ outcompete gaseous CO and several common CO surrogates in this asymmetric process. The reaction mechanism, including the in-situ progressive release of the difluorocarbene, the rapid migratory insertion of $ArPd(II) = CF_2$ species, and subsequent defluorination hydrolysis by water to introduce the carbonyl group, accounts for the overall high efficiency and uniqueness. This work clearly showcases the advantage and potential of the difluorocarbene in synthesis and supplies a mechanistically distinct route for asymmetric carbonylative cyclization reactions.

The difluorocarbene, which can be mildly generated from cheap or easily available difluoromethyl-containing precursors, is a versatile and highly reactive intermediate for organic synthesis[1–5]. Conventionally, it is mainly used to assemble useful *gem*-difluoromethyl molecules through reactions with various nucleophilic substrates, tandem reactions with both a nucleophile and an electrophile, cycloaddition reactions with alkenes or alkynes, or the Wittig reactions with ketones or aldehydes, etc (Fig. 1a)[6–14]. In addition, some unconventional yet highly attractive transformations of the difluorocarbene have recently been disclosed, which enables it a versatile and promising C1 synthon in the synthesis of structurally diverse aza-compounds, aliphatic ethers, and heterocycles in the absence of transition metals[15–20]. However, transition-metal-catalyzed generation of $M = CF_2$ complex from simple precursors and subsequent controllable transformation of it into valuable products remain an unmet need, and limited progress in the palladium difluorocarbene chemistry has been reported until recently. In this regard, Zhang's group synthesized, isolated, and characterized the first example

of $Pd^0 = CF_2$ complex, and developed a controllable Pd-catalyzed difluorocarbene transfer reaction between readily available $BrCF_2PO(OEt)_2$ and arylboronic acids to synthesize difluoromethyl-, tetrafluoroethyl-, difluoroacetyl- or tetrafluoropropanoyl arenes (Fig. 1b)[21–24]. Interestingly, the isolation of carbonyl-incorporated products indicates that the difluorocarbene can serve as a CO surrogate in the presence of $H_2O$[25–28]. Inspired by this work, Song's group reported a Pd-catalyzed assembly of fluoren-9-ones by merging of C-H activation and difluorocarbene transfer, wherein the in-situ formation and subsequent transfer of free CO was proposed[29]. Very recently, they also reported an elegant palladium-catalyzed multi-component difluorocarbene transfer reaction for the selective synthesis of ynones or γ-butenolides (Fig. 1c)[30].

Despite these advances, the metal-catalyzed difluorocarbene transfer is still a very challenging task in terms of the control of reactivity and stereoselectivity due to several factors: (1) the difluorocarbene is a very reactive electrophilic species so that nucleophilic trap of itself will

[1]Shenzhen University of Advanced Technology, Shenzhen 518055, P. R. China. [2]Shenzhen Institute of Advanced Technology, Chinese Academy of Sciences, Shenzhen 518055, P. R. China. [3]School of Chemistry and Chemical Engineering, Nanchang University, Nanchang 330031, P. R. China. [4]Department of Dermatology, State Key Laboratory of Biotherapy and Cancer Center, West China Hospital, Sichuan University, Chengdu 610041, P. R. China. [5]These authors contributed equally: Zhiwen Nie, Keqin Wu, Xiaohang Zhan. ✉e-mail: qin.yin@siat.ac.cn

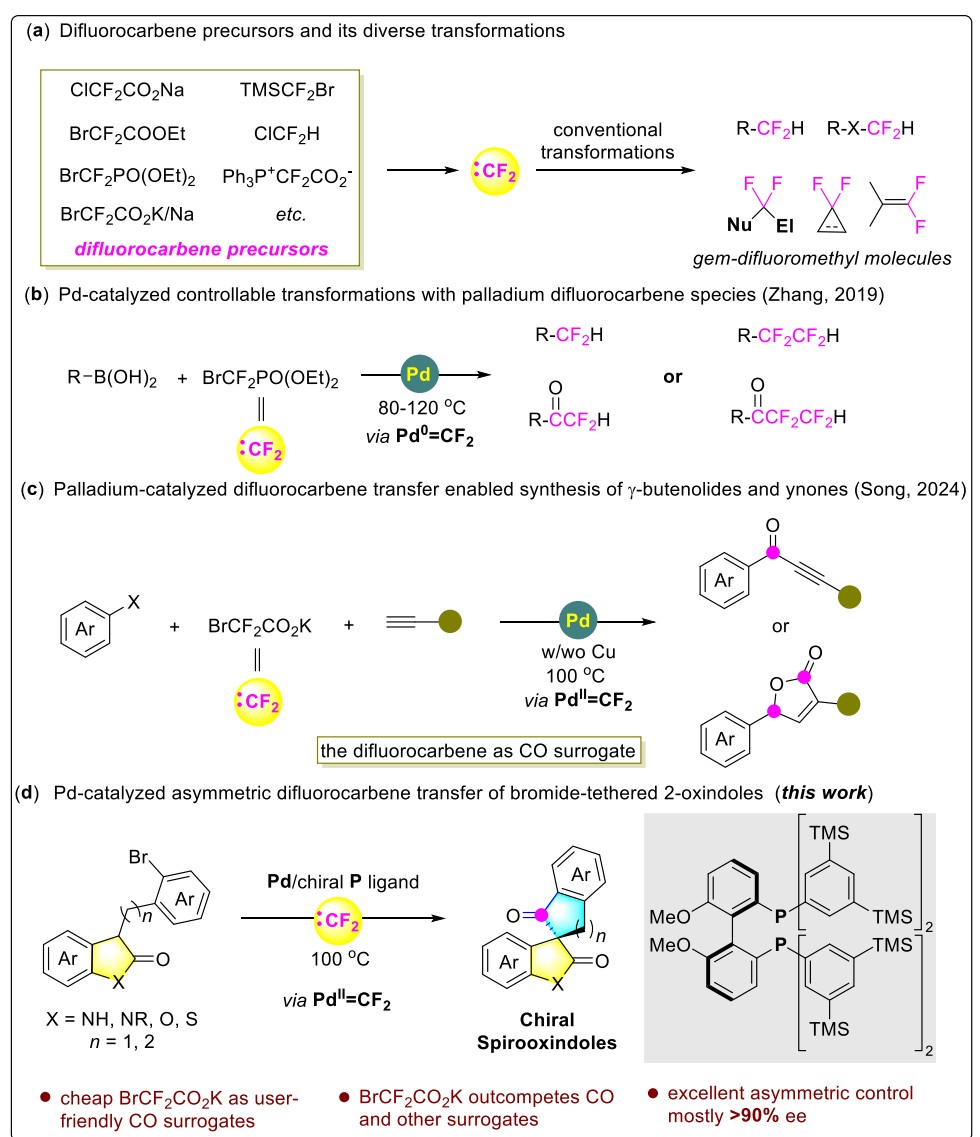

**Fig. 1 | Background of difluorocarbene transfer reaction and our work.**
**a** Difluorocarbene precursors and their conventional transformations. **b** Pd-catalyzed controllable transformations with palladium difluorocarbene species (Zhang, 2019). **c** Palladium-catalyzed difluorocarbene transfer enabling synthesis of γ-butenolides and ynones (Song, 2024). **d** Pd-catalyzed asymmetric difluorocarbene transfer of bromide-tethered 2-oxindoles (this work).

competitively happen prior to the formation of the M = CF$_2$ species, thus affording undesired side products; (2) the high reactivity of the M = CF$_2$ species may also result in uncontrollable reaction pathways, thus increasing the complexity; (3) heating is usually required to release the difluorocarbene from cheap precursors such as BrCF$_2$CO$_2$K/Na, which makes enantiocontrol of the difluorocarbene transfer process very difficult. To our knowledge, difluorocarbene-enabled asymmetric coupling reactions remain elusive and thus highly appealing.

Transition-metal-catalyzed carbonylative cross-coupling reactions of organic halides have now emerged as an efficient method to synthesize versatile ketones in one step[31–33]. In terms of nucleophile types, in situ generated carbanions from α-CH acidic substrates have attracted increasing attention recently due to no necessity for preactivation. This field has been mainly promoted by Wu, Skrydstrup, Beller, and others[34–39]. To be noted, due to the high atomic economy and good compatibility, most examples used CO gas as a C1 synthon. However, the major safety issues associated with toxic gaseous CO cannot be ignored and special equipment such as autoclaves and CO detectors are thus necessary to warrant safe manipulation. To circumvent this problem, chemists have also developed CO surrogates (formic acid derivatives,

metal carbonyls, acyl chloride, and others) for palladium-catalyzed carbonylative reactions[40–47]. The use of weighable and easily handled CO sources has lowered the threshold for reaction equipment and enabled quick advances in carbonylation reactions. However, it is noteworthy that metal-catalyzed asymmetric carbonylation reactions using CO, especially CO surrogates, remain rare and highly challenging[48–56].

On the other hand, chiral spirooxindole skeletons widely exist in natural products, and they are also important synthetic targets[57–59] in that they are often considered pharmacophores for pharmaceutical lead discovery[60]. Catalytic enantioselective synthesis of these attractive structures is of great significance but remains an ongoing challenge[61–65]. To this end, and also inspired by the background that it is feasible to realize a carbonyl group formation from [Pd$^{II}$]= CF$_2$ intermediate with H$_2$O in situ, we decided to explore the Pd-catalyzed difluorocarbene transfer reaction of easily accessible 3-substituted 2-oxindole.

We disclose herein, after extensive investigation, a Pd-catalyzed difluorocarbene transfer reaction to afford a kind of structurally interesting chiral spiro ketones with generally over 90% ee (Fig. 1d). In addition, we also conduct control experiments and mechanistic

## Table 1 | Reaction optimization[a]

| Entry | [Pd] | L | Base | Solvent | Yield (%)[b] | ee[c] |
|---|---|---|---|---|---|---|
| 1 | Pd(OAc)₂ | N-Xantphos | Na₂CO₃ | THF | 97 | 0 |
| 2 | Pd(OAc)₂ | L1 | Na₂CO₃ | THF | 78 | 40 |
| 3 | Pd(OAc)₂ | L2 | Na₂CO₃ | THF | 85 | 31 |
| 4 | Pd(OAc)₂ | L3 | Na₂CO₃ | THF | 88 | 0 |
| 5 | Pd(OAc)₂ | L4 | Na₂CO₃ | THF | 82 | 23 |
| **6** | **Pd(OAc)₂** | **L5** | **Na₂CO₃** | **THF** | **97** | **96** |
| 7 | Pd(OAc)₂ | L6 | Na₂CO₃ | THF | 91 | 67 |
| 8 | Pd(OAc)₂ | L7 | Na₂CO₃ | THF | 83 | 25 |
| 9 | Pd(OAc)₂ | L8 | Na₂CO₃ | THF | 73 | 17 |
| 10 | Pd(PPh₃)₄ | L5 | Na₂CO₃ | THF | 94 | 88 |
| 11 | Pd(OAc)₂ | L5 | K₂CO₃ | THF | 95 | 91 |
| 12 | Pd(OAc)₂ | L5 | NaOH | THF | trace | - |
| 13 | Pd(OAc)₂ | L5 | NaHCO₃ | THF | trace | - |
| 14 | Pd(OAc)₂ | L5 | Na₂CO₃ | dioxane | 82 | 62 |
| 15 | Pd(OAc)₂ | L5 | Na₂CO₃ | toluene | 75 | 34 |
| 16[d] | Pd(OAc)₂ | L5 | Na₂CO₃ | THF | 91 | 96 |
| 17[e] | Pd(OAc)₂ | L5 | Na₂CO₃ | THF | 96 | 96 |
| 18[f] | Pd(OAc)₂ | L5 | Na₂CO₃ | THF | 51 | 96 |

[a]Reaction condition: **1a** (0.1 mmol), BrCF₂CO₂K (2.0 equiv, 0.2 mmol), [Pd] (3.0 mol%), **L** (3.6 mol%), Base (3.0 equiv, 0.3 mmol), solvent (2.5 mL), H₂O (3.5 equiv, 0.35 mmol), 100 °C, 12 h. [b]Isolated yield. [c]Enantiomeric excesses (ee) were determined by HPLC analysis using a chiral stationary phase. [d]BrCF₂CO₂Et instead of BrCF₂CO₂K. [e]TMSCF₂Br instead of BrCF₂CO₂K. [f]Without external H₂O. N-xantphos: 4,6-Bis(diphenylphosphino)phenoxazine.

studies, which exclude the formation and involvement of free CO in the reaction process. Importantly, we also find that BrCF₂CO₂K, the difluorocarbene precursor, outcompete gaseous CO and several common CO surrogates in this asymmetric process, which displays the advantage and potential of the difluorocarbene in versatile organic synthesis.

## Results

### Condition optimization

Aiming to achieve a user-friendly asymmetric carbonylative coupling method and further expand the synthetic utility of difluorocarbene, we decided to investigate the difluorocarbene transfer reaction of the easily prepared oxindole-tethered aryl bromide **1a** under Pd catalysis. Firstly, the non-asymmetric transformation was investigated. Using BrCF₂CO₂K as a difluorocarbene source, Na₂CO₃ as a base, and water as an additive, the reaction proceeded smoothly to provide the desired spirooxindole rac-**2a** with 97% yield in THF in the presence of 3 mol% of Pd(OAc)₂ and N-xantphos (Table 1, entry 1). Indeed, other Pd precursors and achiral bisphosphine ligands were also proved to be efficient (not shown). Encouraged by these results, we further attempted to realize the enantioselective variant since difluorocarbene transfer-enabled asymmetric coupling reactions remain elusive. Firstly, various

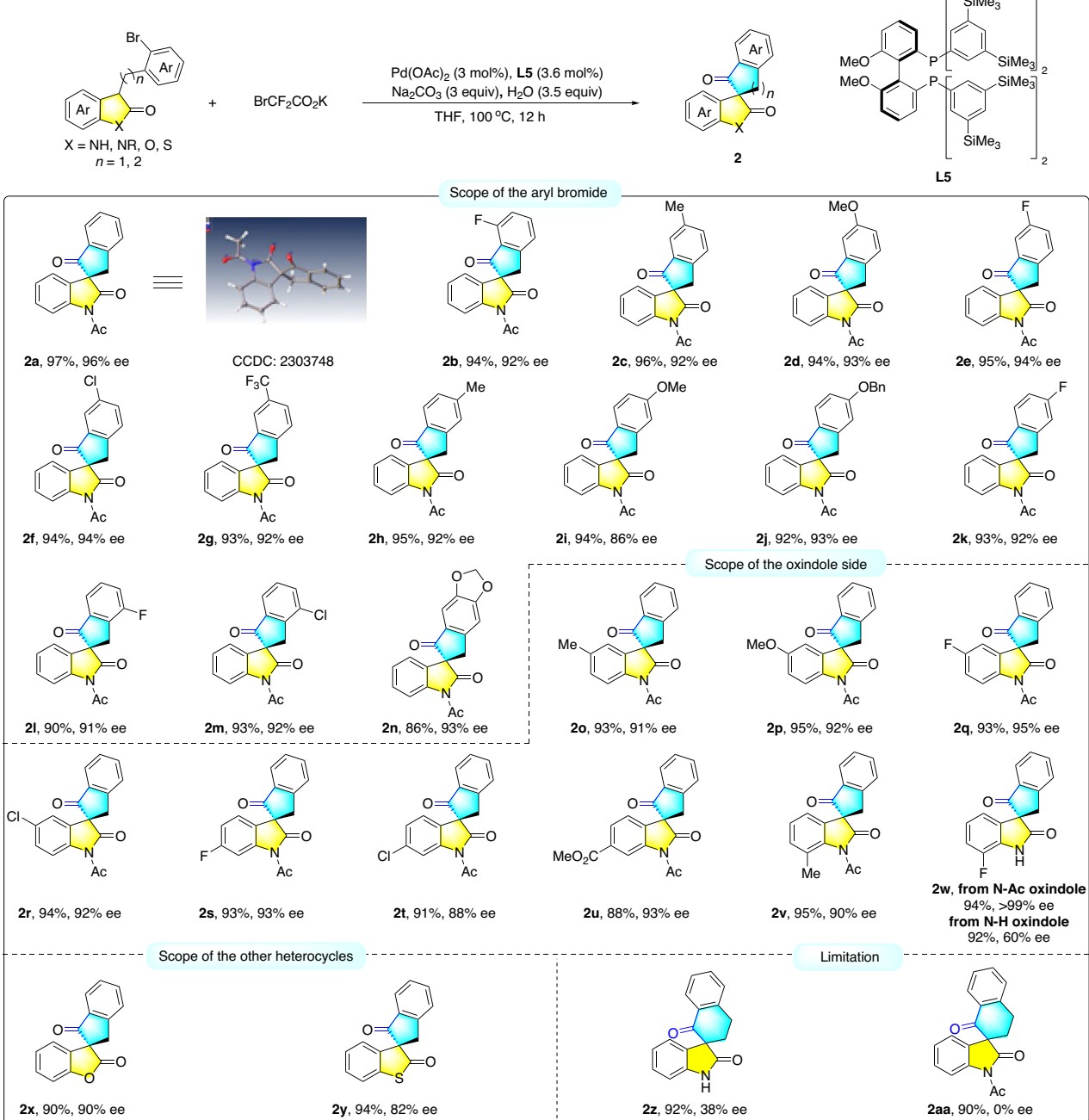

**Fig. 2 | Substrate scope.** Reaction condition: **1** (0.1 mmol), BrCF₂CO₂K (2.0 equiv, 0.2 mmol), Pd(OAc)₂ (3.0 mol%), **L5** (3.6 mol%), Na₂CO₃ (3.0 equiv, 0.3 mmol), THF (2.5 mL), H₂O (3.5 equiv, 0.35 mmol), 100 °C, 12 h. Isolated yield. Enantiomeric excesses (ee) were determined by HPLC analysis using a chiral stationary phase.

chiral bisphosphine ligands were tested, and the ligand **L5** turned out to be the best one (Table 1, entries 2–9). In addition, another Pd source Pd(PPh₃)₄ provided a lower yield and ee value within the indicated time (Table 1, entry 10). It is known that base is critical to the outcome of difluorocarbene-involved transformations as it can affect the releasing rate of difluorocarbene from its precursors, therefore, various bases were next evaluated (Table 1, entries 11–13). When K₂CO₃ was used instead of Na₂CO₃ as a base, the yield and ee value of the desired product **2a** decreased a little bit. In addition, only a trace amount of the desired product **2a** was observed when employing NaOH or NaHCO₃ as a base. The solvent effect is also important to the results as dioxane and toluene led to inferior performance (Table 1, entries 14-15). Other difluorocarbene precursors are also efficient for this transformation, and the desired product **2a** was observed with excellent yield and good

ee value (Table 1, entries 16-17). The external H₂O was also significant to guarantee the high yield of **2a**, as it can accelerate the transformation of difluorocarbene-Pd species to carbonyl-Pd species through the defluorination process, thus driving the reaction ahead (Table 1, entry 18). Therefore, the established optimal conditions for the Pd-catalyzed asymmetric carbonylative reaction were determined as follows, Pd(OAc)₂/**L5** as the catalyst combination, BrCF₂CO₂K and H₂O as the carbonyl source, Na₂CO₃ as the base and THF as the solvent (Table 1, entry 6).

## Substrate scope

With the optimal conditions in hand, a series of oxindole-tethered aryl bromides **1** were prepared and tested to check the substrate generality (Fig. 2). First, the substituent effect of the tethered aryl group was

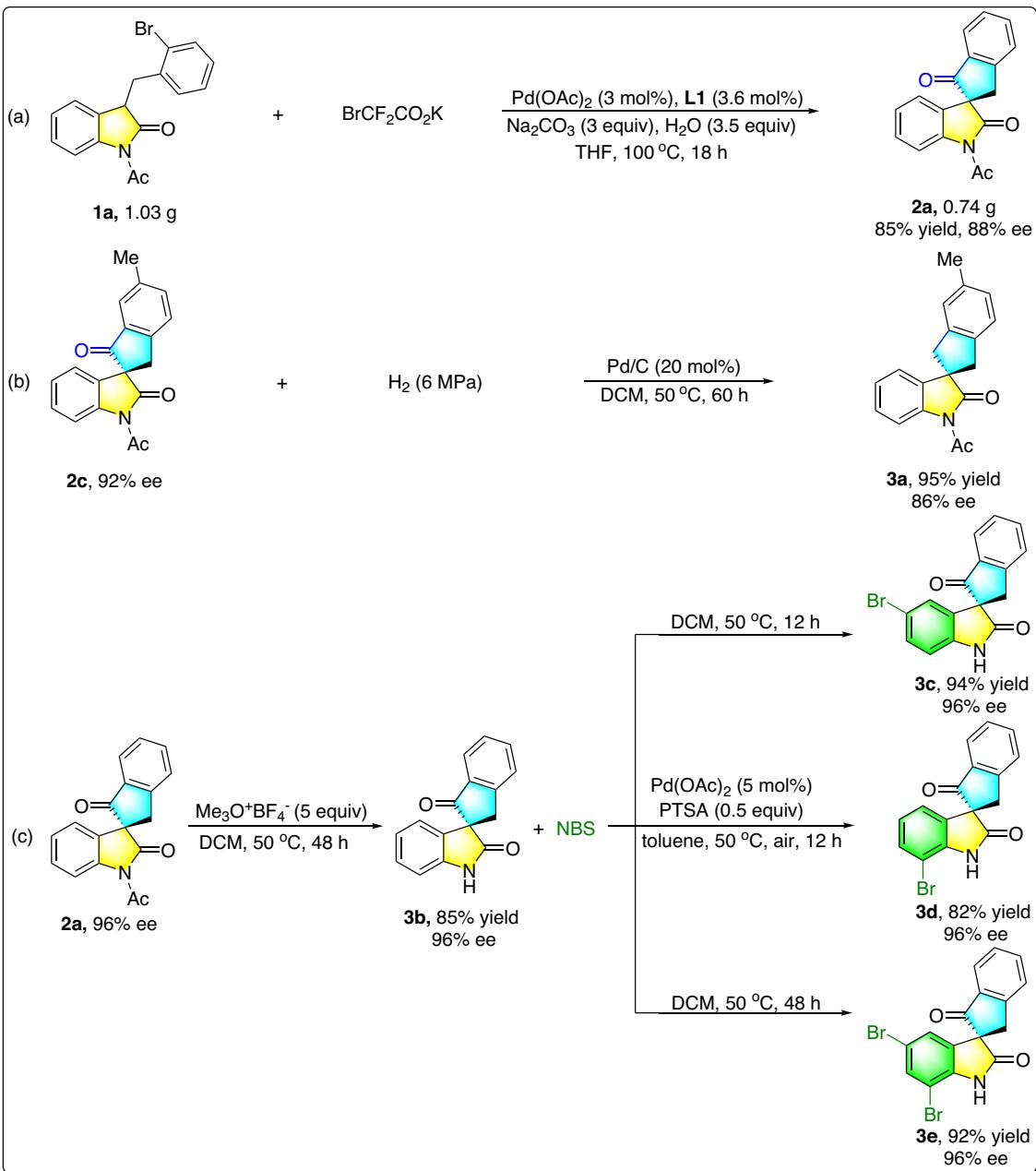

**Fig. 3 | Synthetic applications. a** Gram-scale experiment of (**1a**). **b** Hydrogenation reaction of (**2c**). **c** Transformations of product (**2a**).

evaluated. Overall, electron-donating groups such as Me, MeO, and OBn or electron-withdrawing groups such as Cl, F, and CF₃ on each position of the benzene ring were all well tolerated, and the corresponding products were obtained in excellent yields and enantiocontrol (**2b-2n**, 86%-96% yields, 86%-94% ees). As observed, substituents with diverse properties on various positions of the oxindole aryl part were all compatible, providing the desired products in excellent yields and enantiocontrol (**2o-2v**, 88%-95% yields, 88%-95% ees). Interestingly, N-Ac substituted substrate **1w** underwent a deacylation process to afford the NH spirooxindole **2w** in 94% yield and 99% ee. In comparison, the direct transformation of NH oxindole led to decreased enantioselectivity of **2w** (60% ee), which indicated that the N-deacylation occurred after carbonylative arylation. Subsequently, the substituent effect on the oxindole part was also investigated. Furthermore, the benzo fused lactone **1x** or thiolactone **1y** also worked smoothly to afford the enantioenriched products **2x** and **2y** in 90% ee and 82% ee. The good compatibility of this method allows for a

diversity of chemical space that may be beneficial to drug discovery and development. Unfortunately, the 6-membered cyclic ketone **2z** or **2aa** was attained with only 38% ee or in a racemic form, respectively. Overall, the method exhibits a broad scope and excellent enantiocontrol for various substituents.

## Synthetic applications

To highlight the practicality of this method, a gram-scale asymmetric transformation of **1a** was performed, and the desired product spirooxindole **2a** was obtained in 85% yield and 88% ee (Fig. 3a). Under the catalysis of Pd/C, the hydrogenation reaction of **2c** proceeded smoothly, and the reduced product **3a** was formed without significant loss of enantiopurity (Fig. 3b). In addition, a facile N-deacylation of **2a** had also been conducted, affording the N-unprotected spirooxindole **3b** with complete maintenance of the enantiopurity. Subsequently, regioselective brominations of spirooxindole **3b** were evaluated and the reactions occurred smoothly by regulating the reaction conditions,

**Table 2 | Comparison of various CO sources in asymmetric carbonylative spiroannulation[a]**

| Entry | [CO] | Yield (%)[b] | ee (%)[c] |
|---|---|---|---|
| 1 | CO (balloon) | 0 | - |
| 2 | CO (1.5 MPa) | trace | - |
| 3 | $Mo(CO)_6$ (2.0 equiv) | 0 | - |
| 4 | $W(CO)_6$ (2.0 equiv) | 0 | - |
| 5 | $Co_2(CO)_8$ (2.0 equiv) | 0 | - |
| 6 | $Mn_2(CO)_{10}$ (2.0 equiv) | 0 | - |
| 7 | phenyl formate (2.0 equiv) | 24 | 94 |
| 8[d] | CO (balloon)+ $BrCF_2CO_2K$ | 10 | 0 |
| 9[e] | CO (1.5 MPa) + $BrCF_2CO_2K$ | trace | - |

[a]Reaction condition: **1a** (0.1 mmol), [CO], Pd(OAc)$_2$ (3.0 mol%), **L5** (3.6 mol%), Na$_2$CO$_3$ (3.0 equiv, 0.3 mmol), THF (2.5 mL), 100 °C, 12 h. [b]Isolated yield. [c]enantiomeric excesses (ee) were determined by HPLC analysis using a chiral stationary phase. [d]**1a** (0.1 mmol), BrCF$_2$CO$_2$K (3.0 equiv), CO (balloon), Pd(OAc)$_2$ (3.0 mol%), **L5** (3.6 mol%), Na$_2$CO$_3$ (3.0 equiv, 0.3 mmol), THF (2.5 mL), 100 °C, 12 h. [e]**1a** (0.1 mmol), BrCF$_2$CO$_2$K (3.0 equiv), CO (1.5 MPa), Pd(OAc)$_2$ (3.0 mol%), **L5** (3.6 mol%), Na$_2$CO$_3$ (3.0 equiv, 0.3 mmol), THF (2.5 mL), 100 °C, 12 h.

providing bromo-substituted products **3c, 3d**[66] and **3e** in good yields. To our delight, the enantiopurity of **3c, 3d,** and **3e** were completely maintained (Fig. 3c).

## Mechanistic studies

To better understand the reaction process, we conducted several comparison experiments under standard conditions by using different CO sources (Table 2). To our surprise, BrCF$_2$CO$_2$K displayed much higher reactivity in this asymmetric carbonylative spiroannulation reaction as nearly no reaction (entries 1-6) or low yield (24% yield using phenyl formate, entry 7) was observed when using CO gas or other conventional surrogates. In addition, we had performed the control experiments by adding 1 atm or 1.5 MPa of CO to the standard conditions (entries 8-9). The experimental results show that the existence of free CO gas has detrimental effect on the reaction outcome, and the target product **2a** was obtained with 10% yield or trace, respectively. Particularly, the target product was obtained as a racemic form under 1 atm of CO. These results indicate free CO is probably not involved when using a difluorocarbene precursor, and the excess amount of free CO gas may poison the chiral metal catalyst in this case, likely due to easy ligand substitution of sterically bulky **L5** by CO.

To gain insight into the mechanism, a series of mechanistic experiments were performed (Fig. 4). First, to identify the O source of the newly formed carbonyl group, an O-isotope experiment was carried out. By adding 3.5 equiv of H$_2$$^{18}$O, around 49% of the O atom in the carbonyl group was labeled by mass analysis, which indicated the O atom of the newly formed carbonyl group probably originated from the external water, or trace amount of water in the system or reagents (Fig. 4a). The possibility that the O atom of the newly formed carbonyl group originated from Na$_2$CO$_3$ was excluded since some other organic bases could also promote this transformation (Supplementary Table 12). Next, we experimented with trapping the in-situ-formed difluorocarbene intermediate by adding 2-mercaptopyridine as the trapping reagent under standard conditions. As shown, compound **A** was obtained with 90% yield while the carbonylation product rac-**2a** was not detected (Fig. 4b). **A** was likely generated through the insertion of the S-H bond into the in-situ-formed difluorocarbene, as

evidenced by literature[67]. To further figure out the mechanism, we prepared the palladium complex **B** by treating **1ab** with stoichiometric Pd(PPh$_3$)$_4$, and its structure was determined by NMR and X-ray analysis. When treating **B** with an excess amount of TMSCF$_2$Br and NaOAc, a precursor combination of difluorocarbene[68–70], we observed the formation of a new palladium species bearing a difluoromethyl substituent, which was tentatively identified as the complex **C** following HRMS and $^{19}$F NMR analysis, as well as comparison of documented analogs (Fig. 4c)[71–73]. Attempts to purify and obtain the X-ray structure of **C** were so far unsuccessful. Treatment of complex **C** with Na$_2$CO$_3$ under 100 °C provided the desired product **2ab** quantitatively, indicating **C** is likely in the catalytic cycle (Fig. 4c). However, we failed to observe any palladium species bearing a carbonyl group from this stepwise control experiments. On the other hand, it is also possible that complex **B** first undergoes intramolecular cyclization to afford a 5-membered spiro-Pd-metallic species **E**, followed by insertion of difluorocarbene and subsequent transformations to eventually produce **2ab**. However, no reaction happened at all when complex **B** was treated with Na$_2$CO$_3$ at rt or 100 °C, possibly due to the steric repulsion (Fig. 4c, bottom).

Based on the mechanistic studies and literature precedence[22,74,75], we speculate a possible catalytic cycle as follows (Fig. 5, path 1). First, the Pd(II) complex **I** is formed by the oxidative addition of **1ab**, followed by coordination and facile migratory insertion of a difluorocarbene to afford intermediate **II**. In the presence of base and water, **II** is converted to the palladium species **III**, a process which is evidenced in the literature[71]. The concentration of **II** was maintained at a low level due to progressive generation as well as fast consumption of itself. Subsequently, intramolecular cyclization of **III** assisted by a base affords a 6-membered spiro-Pd-metallic species **IV**, which eventually yields the carbonylative arylation product **2ab** and regenerates the Pd(0) catalyst through reductive elimination. Overall, the reaction is enabled by a difluorocarbene transfer process. This unique reaction process, which does not involve free CO that may poison the chiral metal catalyst, accounts for the overall high efficiency. Another route (Fig. 5, path 2) involving the 5-membered spiro-Pd-metallic species **V** is less likely and tentatively excluded by control experiments.

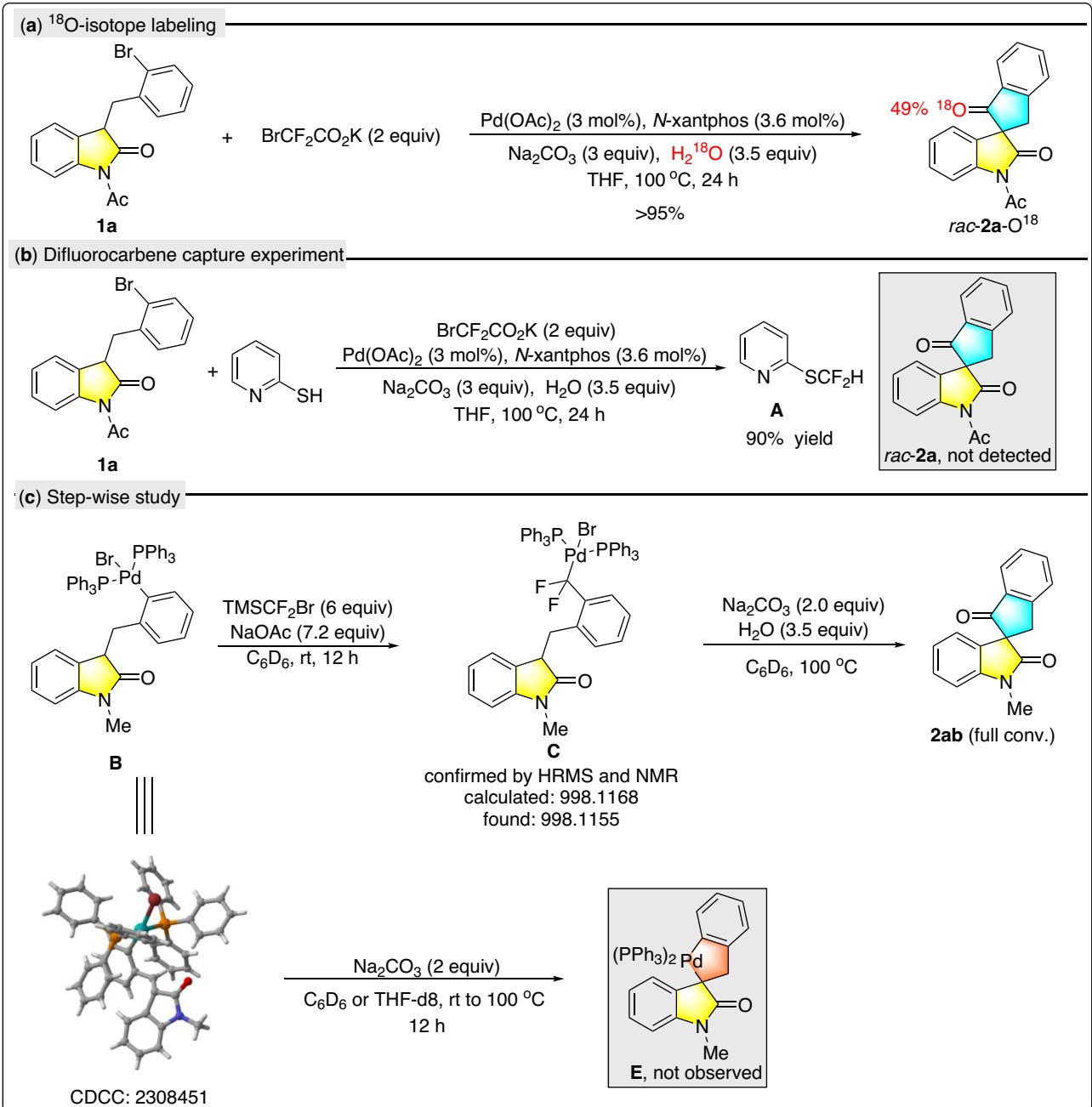

**Fig. 4 | Mechanistic studies. a** [18]O-isotope labeling experiment. **b** Difluorocarbene capture experiment. **c** Step-wise study.

## Discussion

To summarize, in this work we have developed a highly Pd-catalyzed difluorocarbene transfer reaction to afford a broad range of structurally interesting spirooxindoles in excellent yields and ees. Remarkably, difluorocarbene precursors are used as practical, operationally convenient, safe, and efficient CO surrogates in Pd-catalyzed asymmetric cross-coupling reactions. Mechanistic experiments support that the difluorocarbene's insertion into the Pd center occurs before the formation of the spiro-Pd-metallic species. Comparison experiments of various CO sources have shown that difluorocarbene plays multiple roles, not only as an environmentally friendly and safe CO surrogate. We believe this work will stimulate more effort on the challenging asymmetric carbonyl reactions using the difluorocarbene as a CO surrogate.

## Methods

### General procedure for Pd-catalyzed difluorocarbene transfer enables access to chiral spirooxindoles

A reaction tube was charged with $Pd(OAc)_2$ (0.003 mmol, 0.03 equiv), **L5** (0.0036 mmol, 1.2 equiv to [Pd]), **1** (0.1 mmol, 1.0 equiv), $BrCF_2CO_2K$ (0.2 mmol, 2.0 equiv), $Na_2CO_3$ (0.3 mmol, 3.0 equiv), degassed $H_2O$ (0.35 mmol, 3.5 equiv) and THF (2.5 mL) under $N_2$ atmosphere. The reaction vessel was sealed using a PTFE septum, and the mixture was stirred at 100 °C for 12 h. After completion of the reaction, the resulting solution was cooled to room temperature; then it was diluted with DCM (6 mL), washed with water (6 mL), extracted with DCM (6 × 3 mL), and dried over anhydrous $Na_2SO_4$ and concentrated in vacuo. The residue was purified by flash chromatography on silica gel to afford the desired product **2**.

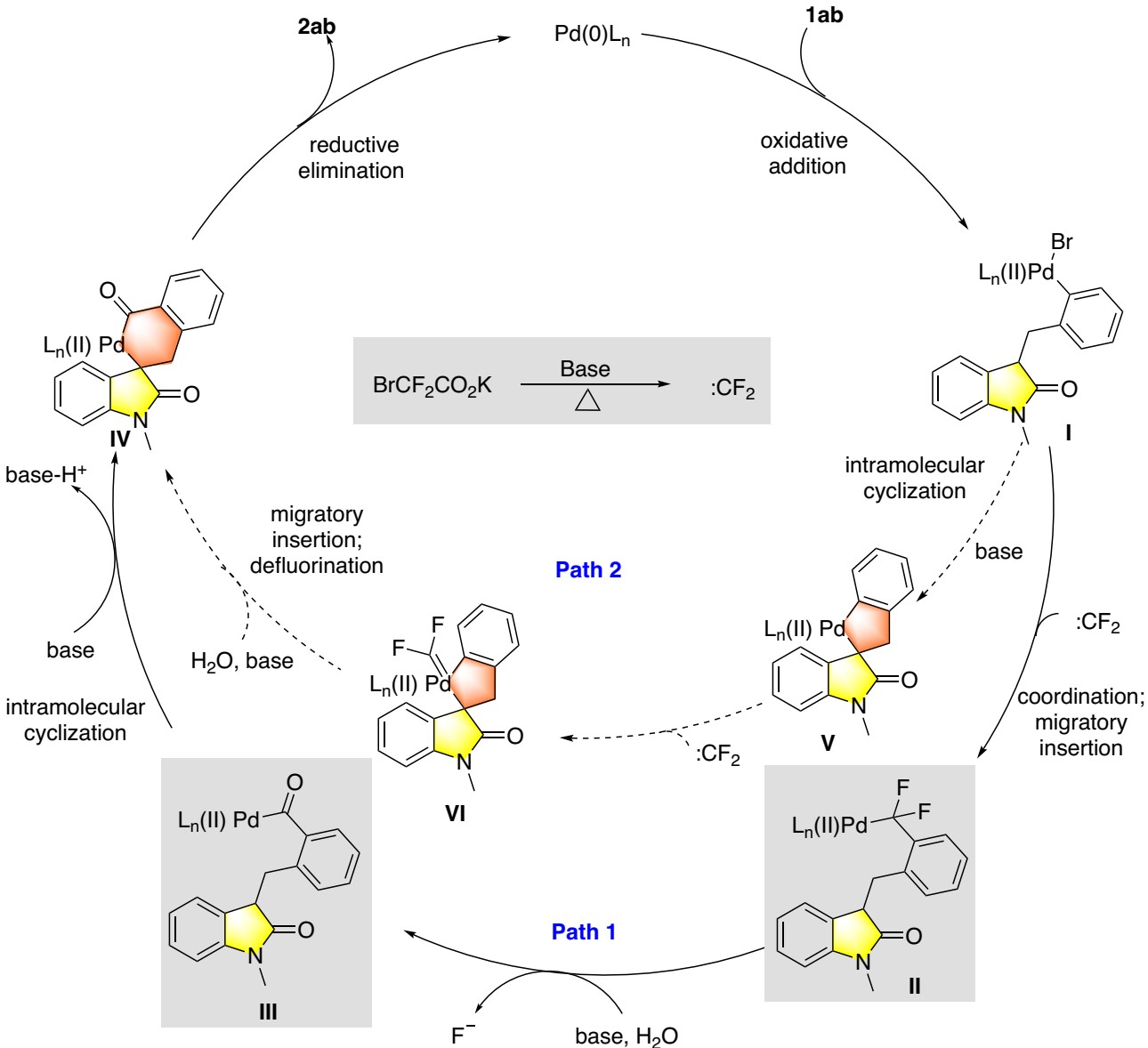

**Fig. 5 | Proposed mechanism. Path 1** the suggested reaction process; **Path 2** an unlikely reaction process.

## Reporting summary

Further information on research design is available in the Nature Portfolio Reporting Summary linked to this article.

## Data availability

The authors declare that the data supporting the findings of this study are available within the paper and its supplementary information files. Data supporting the findings of this manuscript are also available from the authors upon request. Crystallographic data for compounds **2a** and **B** have been deposited in the Cambridge Structural Database with the deposition number 2303748 and 2308451 respectively. Copies of the crystallographic data can be obtained free of charge via https://www.ccdc.cam.ac.uk/structures/.

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

## Acknowledgements

We are grateful to the Guangdong Basic and Applied Basic Research Foundation (2021B1515020062, Q.Y.), and the National Natural Science Foundation of China (Nos. 22071097, Q.Y.; 21991113, Q.Y.). Q.Y. is indebted to Shenzhen University of Advanced Technology, and Shenzhen Institute of Advanced Technology, Chinese Academy of Sciences, for providing a starting grant.

## Author contributions

Z.N., K.W. and X.Z. contributed equally to this work. Z.N. and Q.Y. contributed to the conception and design of the experiments. Q.Y. directed the project. Z.N., K.W. and X.Z. performed the experiments and analyzed the data. W.Y., Z.L., S.L. and S-G.W. provided valuable suggestions to the project. Z.N. and Q.Y. co-wrote the manuscript.

## Competing interests

The authors declare no competing interests.
