## [Peer Review File · Nature Communications]

Palladium-catalyzed asymmetric difluorocarbene transfer enables access to chiral spirooxindolesREVIEWER COMMENTS

Reviewer #1 (Remarks to the Author):

In this work, Yin and coworkers disclosed a very interesting asymmetric carbonylative cyclization reaction using a difluorocarbene precursor as CO surrogate. The difluorocarbene, which can be easily in-situ generated from its precursor, has recently emerged as a versatile synthon for organic synthesis, including serving as an operationally simple surrogate of CO in Pd-catalyzed cross coupling reactions, as properly cited in the manuscript. However, in relevant reports, its role is usually defined as a CO substitute, or as a source of free CO. In this work, the authors discovered unusual results when comparing the performance of the difluorocarbene precursor and CO gas. It's surprising and inspiring to see that BrCF₂CO₂K outcompeted gaseous CO and several common CO surrogates in this asymmetric cyclization process. The authors also carried out mechanistic studies to explain the reaction process, which showcased a mechanistically distinct route for asymmetric carbonylative cyclization reactions without free CO was involved. Mechanistically, the reaction process involves the transfer of a ArPd(II)=CF₂ species, and asymmetric difluorocarbene transfer reactions have never been reported prior to this work. The reaction works very well for a broad range of substrates, generally with excellent yields and ee, and the chiral spirooxindole products are a kind of privileged scaffold for drug discovery. In addition, the manuscript is well written and the SI is well prepared.

Based on the novelty and significance of this work, I will recommend the acceptance of this work after minor revision.

Some concerns are listed below:

1. As the author argue that free CO may poison the chiral catalyst, this reviewer is curious about the results of the comparison experiments by adding 1 atm or higher pressure of CO to the standard conditions (in the presence of difluorocarbene precursor). The results might be indicative.
2. "N" of N-Xantphos should be italic throughout the manuscript.

3. the definition of “BINAP: 1.1'-Binaphthyl-2.2'-diphemyl phosphine.” In the footnote of Table 1 is redundant as its structure is depicted already.

4. In the supporting information, it should be mentioned from which provider the chiral ligands were purchased, or if they were synthesised by the authors. In the latter case, full procedures and characterization data should be provided. Meanwhile, the supplier of the optimal base Na₂CO₃ should also be added.

Reviewer #2 (Remarks to the Author):

Yin et al reported an intriguing work to construct a board range of biologically interesting enantioenriched spirooxindoles via Pd-catalyzed asymmetric difluorocarbene transfer strategy. The combination of BrCF₂CO₂K and water as a precursor of CO in this carbonyl cyclization process is remarkable, quite different from the conventional route with the requirement of CO or related, especially in enantioselective synthesis. Surprisingly, several other CO surrogates do not show competing efficiency in this asymmetric process. The authors also conduct a series of mechanistic experiments to support a distinct reaction route, which involves generation of the difluorocarbene and following ArPd(II)=CF₂ species, as well as subsequent migratory insertion and defluorination hydrolysis by water. Therefore, the overall reaction process is different from reported cases using free CO. Thus, I recommend the acceptance of this work. However, there are also minor suggestions for the improvement of this work.

(1) The authors described that “the excess amount of free CO gas may poison the chiral metal catalyst in this case, likely due to easy ligand substitution of sterically bulky L5 by CO.” If this happens, the ee of the product should vary as well. What is the ee of the product (trace but isolable?) in the presence of 1.5 MPa CO?

(2) Following the first question, what is the results if gaseous CO is added under the standard conditions? These results could be informative.

(3) Regarding the reaction mechanism, it seems that the final cyclization (possibly the C-C formation step) might be the rate-determining step, due to the adoption of high reaction temperature for this step in Fig 4c. However, the defluorination step (from II to III in Fig 5) is

also involved inside. To figure out the actual RDS, a KIE test with the use of deuterated H₂O is suggested to provide more mechanistic information.

(4) For the SI, the peak intensity of some ¹H NMR and ¹⁹F NMR spectra should be increased.

Reviewer #3 (Remarks to the Author):

The catalytic enantioselective synthesis of chiral spirooxindole skeleton structures is of great significance for organic and pharmaceutical synthesis. Inspired by previous literature about the formation of carbonyl group from [PdII]=CF₂ intermediate with H₂O in situ, Yin et al. developed a highly enantioselective Pd-catalyzed difluorocarbene transfer reaction to afford a broad range of structurally interesting spirooxindoles in excellent yields and ees. The authors also performed control experiments to support the proposed reaction mechanism. Although the study of using difluorocarbene precursors as a carbonyl source has been reported in many references, considering that metal-catalyzed asymmetric carbonylation reactions using CO surrogates remain rare and highly challenging as mentioned in the introduction of this manuscript, I recommend accepting the publication of this work after completing the following minor revisions.

(1) Some new references about difluorocarbene as a carbonyl source would better be cited, such as *Org. Biomol. Chem.*, 2022,20, 8120-8124; *Eur. J. Org. Chem.* 2024, e202400032.

(2) In Figure 2, preparing the target product 2W from N-H oxindole shows less ee (60%) than from N-Ac oxindole (>99% ee), why? Authors should try to explain it.

(3) In Figure 2, products 2z and 2aa show low (38%) or no (0%) ee, why?

(4) Figure 4a, to verify the origin of the carbonyl oxygen atom in the product, the authors performed O-isotope experiments, around 49% of the O atom in the carbonyl group was labeled by mass analysis. However, the content of ¹⁸O is not ideally high. It might be the competitive reactions between H₂O from reaction system and H₂¹⁸O when conducting O-isotope experiments. Could the reaction be strictly controlled? Whether other compounds, including reaction substrates, bases, and reaction solvents have been dried carefully. If

not, further research should be done. In addition, attention should be paid to the writing of details, H₂O¹⁸ should be changed to H²¹⁸O.

(5) Figure 4c, Pd complex B undergoes a subsequent reaction to obtain the target product 2ab. Why didn't use THF as the solvent but C₆D₆? In addition, if THF is used in the process from B to E, can intermediate E be detected?

(6) Please carefully check characterization data and NMR spectra in Supporting Information, especially for fluorinated compounds. For example, 2q shows more peaks in its ¹³C NMR spectrum than spectra data; and 2r and 2x show high solvent peaks in ¹H NMR spectra. The spectra should be improved after further product purification.

Thank you for your email on 07-June-2024. We appreciate very much your consideration of our manuscript (Manuscript number: NCOMMS-24-29793) and the valuable comments and suggestions from the reviewers. The quality of this work undoubtedly benefits a lot from the suggestions. We have addressed these without exception point-by-point in this response letter. Please find enclosed the revised manuscript including a highlighted version and the amended Supporting Information.

Please find our point-by-point response listed below:

Comments from reviewer 1

1. As the author argue that free CO may poison the chiral catalyst, this reviewer is curious about the results of the comparison experiments by adding 1 atm or higher pressure of CO to the standard conditions (in the presence of difluorocarbene precursor). The results might be indicative.

Response: Agree and appreciated for the suggestions.

We have performed the comparison experiments by adding 1 atm or 1.5 MPa of CO to the standard conditions (in the presence of BrCF₂CO₂K). The experimental results show that the existence of free CO gas has detrimental effect on the reaction outcome, and the target product **2a** was obtained with 10% yield or trace, respectively. Particularly, the target product was obtained as a racemic form under 1 atm of CO. These results clearly support free CO poisons the chiral catalyst.

We updated the data and added the discussion in the revised manuscript.

Entry	CO	Yield (%)	ee (%)
1	CO (balloon)	10	0
2	CO (1.5 MPa)	trace	–

2. “N” of N-Xantphos should be italic throughout the manuscript.

Response: Thank you for your reminder, we have italicized the N atom in N-Xantphos throughout the manuscript.

0. the definition of “BINAP: 1.1'-Binaphthyl-2.2'-diphemyl phosphine.” In the footnote of Table 1 is redundant as its structure is depicted already.

Response: Agree. We have deleted the definition of “BINAP: 1.1'-Binaphthyl-2.2'-

diphemyl phosphine.” in the footnote of Table 1.

4. In the supporting information, it should be mentioned from which provider the chiral ligands were purchased, or if they were synthesised by the authors. In the latter case, full procedures and characterization data should be provided. Meanwhile, the supplier of the optimal base Na₂CO₃ should also be added.

Response: Agree. All ligands were purchased from Bide Pharmatech (a Chinese supplier) and we have made corresponding instructions in the supporting information (part 1: general information). In addition, we have also evaluated the effect of Na₂CO₃ from various suppliers, and the results show that the Na₂CO₃ supplied by Bide Pharmatech provided the best outcome. The results were listed in table S4 in SI.

Entry	Na ₂ CO ₃ supplier	Specification	Content (%)	Yield (%)	ee (%)
1	Adamas	2.5 kg	≥99.5	83	96
2	Bide	500 g	≥99	97	96
3	Leyan	100 g	99.95	85	92
4	J&K	5 g	≥99	95	94
5	TCI	5 g	≥99.5	90	95

Comments from reviewer 2

1. The authors described that “the excess amount of free CO gas may poison the chiral metal catalyst in this case, likely due to easy ligand substitution of sterically bulky L5 by CO.” If this happens, the ee of the product should vary as well. What is the ee of the product (trace but isolable?) in the presence of 1.5 MPa CO?

Response:

Please see the reply to comment 1 from the reviewer 1.

2. Following the first question, what is the results if gaseous CO is added under the standard conditions? These results could be informative.

Response: Please see the reply to comment 1 from the reviewer 1.

3. Regarding the reaction mechanism, it seems that the final cyclization (possibly the C-C formation step) might be the rate-determining step, due to the adoption of high reaction temperature for this step in Fig 4c. However, the defluorination step (from II

to III in Fig 5) is also involved inside. To figure out the actual RDS, a KIE test with the use of deuterated H₂O is suggested to provide more mechanistic information.

Response:

Response: Thanks for the suggestion. Accordingly, we conducted comparison experiments using H₂O or D₂O, and the results showed that no difference was observed on the yield of **2a** within either 6 h or 12 h. The results suggest the defluorination step is possibly not the RDS. Additionally, the work by Hu and Gao (Angew. Chem. Int. Ed. 2022, e202213646) showed that the formation of carbonyl group via water-assisted defluorination of the ArPd(II)=CF₂ species could easily proceed at 60 °C.

4. For the SI, the peak intensity of some ¹H NMR and ¹⁹F NMR spectra should be increased.

Response: Agree.

Thank you for your reminder, we have increased the peak intensity of some ¹H NMR and ¹⁹F NMR spectra, such as those of compounds **2b**, **2d**, **2q**, **2r** and **2x**.

Comments from reviewer 3

1. (1)Some new references about difluorocarbene as a carbonyl source would better be cited, such as Org. Biomol. Chem., 2022,20, 8120-8124; Eur. J. Org. Chem. 2024, e202400032.

Response: Agree.

We have added the reference in the revised manuscript, as ref 27 and 28.

2. In Figure 2, preparing the target product **2W** from N-H oxindole shows less ee (60%) than from N-Ac oxindole (>99% ee), why? Authors should try to explain it.

Response: Actually, the N-substituent has obvious effect on the enantiocontrol, as showed in Table S10. The reaction of N-H oxindole delivered the corresponding product in much lower ee than that from N-Ac oxindole.

For the asymmetric transformation of substrate N-Ac-**1w**, we speculated a two-step process, including asymmetric cyclization and subsequent de-acylation, was involved. Therefore, with the N-Ac substrate, >99% ee of **2w** was obtained. However, only 60% ee was obtained when directly using NH-**1w**.

The discussion was already described in the manuscript.

3. In Figure 2, products **2z** and **2aa** show low (38%) or no (0%) ee, why?

Response: In this work, the size of ring produces obvious effect on the enantiocontrol, and only 5-membered ring products can gain excellent control. We repeated these results for several times. We tentatively speculate the ring conformation of various size leads to tremendous energy difference during the enantio-determining step (the cyclization step).

4. Figure 4a, to verify the origin of the carbonyl oxygen atom in the product, the authors performed O-isotope experiments, around 49% of the O atom in the carbonyl group

was labeled by mass analysis. However, the content of ^{18}O is not ideally high. It might be the competitive reactions between H_2O from reaction system and H_2O^{18} when conducting O-isotope experiments. Could the reaction be strictly controlled? Whether other compounds, including reaction substrates, bases, and reaction solvents have been dried carefully. If not, further research should be done. In addition, attention should be paid to the writing of details, H_2O^{18} should be changed to H_2^{18}O .

Response: Thank you for your comment. In previous experiments, we have dried reaction substrates, bases, catalysts, ligands and reaction solvents carefully. However, it seems really hard to prevent the effect of trace amount of water from the reaction system. Even with these efforts, the experimental results show that the content of ^{18}O in the product is not ideally high. Subsequently, we added a verification experiment, that is, using TMSCF_2Br (ideally, containing no water), instead of $\text{BrCF}_2\text{CO}_2\text{K}$, as the source of difluocarbene, Cs_2CO_3 (99.99% purity) as the base, and $\text{Pd}(\text{PPh}_3)_4$ as the catalyst, and the reaction was carried out in dry THF in the glovebox. Although all raw materials in this reaction were rigorously dried, experimental results show that only 50% of the O atoms in the carbonyl group was labeled by mass spectrometry.

In addition, according to suggestion, we have made corresponding changes in the revised manuscript and SI by replacing H_2O^{18} with H_2^{18}O .

M/Z	Abundance	M/Z	Abundance	M/Z	Abundance	M/Z	Abundance
35	1220	101	9726	171	711	234	5314
36	6258	102	16366	172	801	235	3667
37	766	103	11223	173	778	236	1198
38	4265	104	7453	174	2281	237	898
39	14864	105	9442	175	7711	238	414
40	3087	106	3186	176	40550	239	1203
41	13937	107	3075	177	38662	240	481
42	6473	108	1531	178	29646	241	716
43	131650	109	2852	179	18575	242	321
43	17817	110	7803	180	3535	243	481
45	13503	111	5107	181	1121	244	599
46	452	112	6139	182	511	245	714
47	874	113	16000	183	1162	246	888
48	182	114	8478	184	511	247	1021
49	1950	115	19988	185	716	248	67153
50	11977	116	10465	186	643	249	370953

51	21749	117	7996	187	852	250	105484
52	5025	118	2484	188	4008	251	232714
53	3072	119	7476	189	6524	252	39647
54	1686	120	1686	190	52394	253	8434
55	12194	121	4382	191	49150	254	1318
56	5561	122	2818	192	35908	255	1004
57	32009	123	1985	193	68974	256	481
58	2512	124	2071	194	11106	257	777
59	2496	125	5099	195	3063	258	550
60	2332	126	8646	196	893	259	524
61	2089	127	6703	197	884	260	391
62	7894	128	6607	198	585	261	643
63	25913	129	3844	199	953	262	885
64	7188	130	6462	200	409	263	6806
65	6912	131	3986	201	4386	264	1506
66	1050	132	4970	202	41410	265	3415
67	2656	133	9036	203	19807	266	1169
68	1863	134	2323	204	54068	267	6018
69	8950	135	8371	205	24134	268	2039
70	14764	136	2124	206	54833	269	2356
71	19428	137	5234	207	59769	270	789
72	1951	138	4407	208	12202	271	937
73	30846	139	15596	209	8414	272	471
74	10010	140	8766	210	1758	273	425
75	21323	141	3394	211	1791	274	236
76	31863	142	847	212	604	275	558
77	30951	143	1540	213	1673	276	209
78	5660	144	28541	214	586	277	574
79	3961	145	4826	215	705	278	278
80	1087	146	2163	216	1569	279	14538
81	7562	147	9686	217	1828	280	2936
82	13444	148	2402	218	1522	281	18934
83	9506	149	64566	219	18009	282	5713
84	8528	150	16076	220	732618	283	4548
85	4151	151	19876	221	181169	284	2223
86	4739	152	15369	222	43842	285	924
87	10313	153	4519	223	9884	286	485
88	16392	154	1889	224	1673	287	348
89	46638	155	1932	225	1230	288	119
90	29481	156	1163	226	721	289	372
91	12458	157	1570	227	806	290	418
92	1887	158	814	228	935	291 (M)	38664
93	2670	159	1502	229	625	292	10039

94	1155	160	669	230	6652	293 (M+2)	38874
95	4211	161	4449	231	7327	294	6861
96	12541	162	4193	232	11735	295	4461
97	5273	163	27388	233	6818	296	1231

$I(M) : I(M+2) = 38664 : 38874 = 1 : 0.995$; ^{18}O enrichment = $0.995/1.995 = 50\%$

5. Figure 4c, Pd complex **B** undergoes a subsequent reaction to obtain the target product **2ab**. Why didn't use THF as the solvent but C_6D_6 ? In addition, if THF is used in the process from **B** to **E**, can intermediate **E** be detected?

Response: Thank you for your comment. First, the use of C_6D_6 as the solvent was due to the reason that the target product could be obtained with higher yield. Secondly, the intermediate **C** can be better characterized by NMR using C_6D_6 as the solvent.

According to suggestions, we studied the process of **B** to **E** using THF-d8 as the solvent, however, ^{31}P NMR and HRMS analysis showed that the intermediate **E** was not observed, perhaps the process was prevented by the bulky steric hindrance since the formation of spiro skeleton was unfavorable.

6. Please carefully check characterization data and NMR spectra in Supporting Information, especially for fluorinated compounds. For example, **2q** shows more peaks in its ^{13}C NMR spectrum than spectra data; and **2r** and **2x** show high solvent peaks in ^1H NMR spectra. The spectra should be improved after further product purification.

Response: Thank you for your reminder, we have carefully checked the characterization data and NMR spectra in Supporting Information. Compounds **2q**, **2r** and **2x** were further purified and re-characterized by NMR and made the corresponding corrections in the revised Supporting Information.

We hope that we have the reviewers' comments carefully and adequately addressed and thank you for considering our revised manuscript for acceptance in *Nature Communications*.

Many thanks,

Qin Yin

REVIEWERS' COMMENTS

Reviewer #1 (Remarks to the Author):

All the issues have been addressed. I am glad to accept this manuscript.

Reviewer #2 (Remarks to the Author):

The authors have addressed the previous issues and now I believe this manuscript should be suitable for publication.

Reviewer #3 (Remarks to the Author):

The authors described a Pd-catalyzed enantioselective difluorocarbene transfer reaction to access a series of chiral spiro ketones with generally over 90% ee. Mechanistic studies showed that difluorocarbene in situ formed from $\text{BrCF}_2\text{CO}_2\text{K}$ hydrolyzed to carbonyl group after coordination and migratory insertion. In the revised manuscript, the authors have corrected and revised all mistakes and problems proposed by reviewers, which makes the work comprehensive and credible. Now I recommend it be published in Nature Communications.